

# Aerosol models from the AERONET data base. Application to surface reflectance validation

Jean-Claude Roger[1,2], Eric Vermote[2], Sergii Skakun[1,2], Emilie Murphy[1,2], Oleg Dubovik[4], Natacha Kalecinski[1,2], Bruno Korgo[3], Christopher Justice[1], and Brent Holben[5]

[1] Department of Geographical Sciences, University of Maryland, College Park, MD, 20742, USA
[2] Terrestrial Information System Branch-Code 619, NASA/GSFC, Greenbelt, MD, 20771, USA
[3] Laboratoire d'Optique Atmosphérique, Université de Lille 1, Villeneuve d'Ascq, 59665, France
[4] Laboratory of Thermal and Renewable Energy, University Joseph KI-ZERBO, Ouagadougou, Burkina Faso
[5] Biospheric Science Branch-Code 618, NASA/GSFC, Greenbelt, MD, 20771, USA

*Correspondence to*: Jean-Claude Roger (roger63@umd.edu)

**Abstract.** Aerosols play a critical role in radiative transfer within the atmosphere, and they have a significant impact on climate change. As part of the validation of atmospheric correction of remote sensing data affected by the atmosphere, it is critical to utilize appropriate aerosol models as aerosols are a main source of error. In this paper, we propose and demonstrate a

framework for building and identifying an aerosol model. For this purpose, we define the aerosol model by recalculating the aerosol microphysical properties ($C_{vf}$, $C_{vc}$, $\%C_{vf}$, $\%C_{vc}$, $r_{vf}$, $r_{vc}$, $\sigma_r$, $\sigma_c$, $nr_{440}$, $nr_{650}$, $nr_{850}$, $nr_{1020}$, $ni_{440}$, $ni_{650}$, $ni_{850}$, $ni_{1020}$, $\%S_{ph}$) based on the optical thickness at 440 nm $\tau_{440}$ and the *Ångström* coefficient $\alpha_{440-870}$ obtained from numerous AERosol RObotic NETwork (AERONET) sites. Using aerosol microphysical properties provided by the AERONET dataset, we were able to evaluate our own retrieved microphysical properties. The associated uncertainties are up to 23%, except for the challenging,

imaginary part of the refractive index $ni$ (about 38%). Uncertainties of the retrieved aerosol microphysical properties were incorporated in the framework for validating surface reflectance derived from space-borne Earth observation sensors. Results indicate that the impact of aerosol microphysical properties varies $3.5\times10^{-5}$ to $10^{-3}$ in reflectance units. Finally, the uncertainties of the microphysical properties yielded an overall uncertainty of approximately of 1 to 3% of the retrieved surface reflectance in the MODIS red spectral band (620-670 nm), which corresponds to the specification used for atmospheric correction.

# 1 Introduction

Aerosols play a key role in the atmosphere as an important forcer in climate assessment (IPCC, 2018; IPCC, 2019) and we definitively need to improve our knowledge of their properties for a better evaluation of their impacts (Dubovik and King, 2000; Andreaa et al., 2002; Dubovik et al., 2002b; Roger et al., 2009; Omar et al, 2008; Nousiainen, 2011; Dubovik et al., 2011; Ginoux et al., 2012; Boucher et al. 2013; Calvo et al., 2013; Lenoble et al., 2013; Fuzzi et al. 2015; Derimian et al.,

2016; Klimont et al., 2017; Torres et al., 2017; Bond et al., 2018; Contini et al., 2018; De Sá et al. 2019; Li et al., 2019, Mallet et al., 2020…).





In general, the study or use of an aerosol model needs careful consideration. While a rough description of an aerosol model is sometimes adequate (e.g., when undertaking a long-term study whereby a mean description of aerosols is enough, because their properties might vary considerably in time and in space, or simply when aerosols don't have a specific impact on the

radiative transfer or on the climate), other studies require a more specific and precise characterization. One such area of study is the validation of surface reflectance products derived from the space-borne sensors, which was a major rationale for NASA supporting the AERONET network (Holben et al.,1998).

In this context, and through the lens of satellite product validation, the surface reflectance retrieval requires a good characterization of the aerosol properties, particularly for sensors with various and narrow spectral bands (Justice et al, 2013).

Indeed, by comparing the inverted surface reflectance to the reference surface reflectance, we may be able to validate the surface reflectance product. It is essential, in that case, that a careful validation be performed on a global and continuous basis, including a wide range of land conditions. One approach is the direct comparison with 'ground- truth' measurements, but that presents several challenges related to the scale and nature of the ground measurements and their representativeness at coarse and medium satellite pixel resolutions, since the global representativeness of the pixel may differ from the point measurements.

Nevertheless, at a finer spatial resolution (pixels less than 30m), different possibilities of direct measurement occur. Indeed, with a good protocol and good radiometry, direct ground truth measurements can be performed for validation (Helder and al., 2012; Czapla-Myers et al., 2015; Czapla-Myers et al., 2016; Badawi et al., 2019; Bouvet et al., 2019). There are also other approaches. For example, as alternative, we use an indirect approach for the validation of the official products of MODIS and VIIRS (Vermote et al., 2002; Vermote et al., 2014) and for the NASA Harmonized Landsat-8/Sentinel-2 project (Vermote et

al., 2016; Claverie et al., 2018; Doxani et al., 2018). We compare a retrieved surface reflectance to a surface reflectance reference determined from the Top of Atmosphere (TOA) reflectance corrected using the accurate radiative transfer 6SV code (Vermote et al., 1997; Kotchenova et al., 2006; Kotchenova et al., 2007; Kotchenova et al., 2008) and detailed measurements of the atmosphere. An intermediate step consists in validating the Aerosol Optical Thickness, which is further used as an input to the atmospheric correction process. Numerous references address this, including those used applied to MISR, MODIS,

POLDER and Landsat data (e.g. Martonchik et al., 1998; Remer et al., 2005; Herman et al., 2005; Masek et al, 2006; Keller et al., 2007; Martonchik et al., 2009; Dubovik et al., 2011; Levy et al, 2013; Vermote et al, 2016; Levy et al., 2018; Doxani et al., 2018).

As important inputs for atmospheric correction, we need the aerosol properties exactly when the satellite overpasses one of the AERONET validation sites. For the purpose of the surface reflectance product validation, we decided to create a dynamic

aerosol model for each AERONET site when all data are sufficiently available and representative. This paper describes how we define and design these aerosol models for the 850 AERONET selected sites (here the optical model is defined by the microphysical properties). These models can then be used for the atmospheric correction validation, as we do here, or for other purposes when an aerosol model is needed (e.g. local studies or creation of an aerosol climatology).



## 2 Description of the aerosol model

### 2.1 Aerosol microphysical description

The aerosol optical properties (scattering and absorbing coefficients; phase matrix) are derived from the three microphysical properties which define the aerosol model: the size-distribution (gives the diameter distribution of the aerosol population), the complex refractive index (gives the path of light through the atmosphere), and the sphericity (describes the aerosol shape and non-sphericity aspect) (Hansen and Travis, 1974; Van der Hulst, 1981; Lenoble, 1993; Liou, 2002; Mishchenko et al., 2002;

Bohren and Huffman, 2010; Lenoble et al., 2013).

The size distribution characterization may be variable in its chemical or optical description i.e., mass and numbers respectively. This results in a different shape and description of the size-distribution. For an optical approach, the use of the Gaussian Distribution is widely accepted. Thus, to design an optical aerosol size-distribution in its vertical description, a combination of a Gaussian's law for each aerosol mode is suitable (the fine mode and the coarse mode identified hereafter by $f$ and $c$), even

if it can be much more complex at a local state (Liou, 2002; Hsu et al., 2004; Roger et al., 2009; Dubovik and King, 2000; Dubovik et al., 2011; Lee et al., 2015). In this way, the particle volume size-distribution can be described by the derivative of the particle volume at a specific radius V(r) by the natural logarithm of the radius:

$$\frac{dV(r)}{dlnr} = \frac{C_{vf}}{\sqrt{2\pi}\sigma_f} exp\left[-\frac{(lnr-ln\overline{r_{vf}})^2}{2\sigma_f^2}\right] + \frac{C_{vc}}{\sqrt{2\pi}\sigma_c} exp\left[-\frac{(lnr-ln\overline{r_{vc}})^2}{2\sigma_c^2}\right], \qquad (1)$$

where the 6 microphysical parameters that described this model are: $C_{vf}$ (the particle volume concentration of the fine mode),

$C_{vc}$ (the particle volume concentration of the coarse mode), $\overline{r_{vf}}$ et $\overline{r_{vc}}$ (the particle median volume radius of the fine and coarse mode), $\sigma_f$ and $\sigma_c$ (the standard deviation of the Gaussian's law of the fine and coarse mode).

In theory, and most of the time, the phase function of aerosols is normalized (Lenoble, 1985), thus the size distribution doesn't need to be defined in an absolute manner. We then may define the relative volume concentration $\%C_{vf}$ and $\%C_{vc}$ (scaled between 0 and 1), rather than $C_{vf}$ and $C_{vc}$ (discussed latter in this paper). The complex refractive index of the aerosol, $n = n_r +$

$i\ n_i$, is the second required microphysical parameter. The real part ($n_r$) describes the scattering properties of the aerosol, while the imaginary one ($n_i$) describes absorption properties. Both parts have to be known for a given wavelength. Finally, the percentage of sphericity $\%S_{ph}$ can be considered as well, to take into account the nonsphericity of aerosols (Mishchenko et al., 2000; Dubovik et al., 2002b; Herman et al., 2005;), in contrast to a "spherical approach" (Mie, 1908).

### 2.2 Description of the dataset

Aerosol microphysical properties data were extracted from the AERONET network measurements (Holben et al., 1998; Dubovik and King, 2000; Dubovik et al., 2000; Sinyuk et al., 2007). We used the Level 2.0 (quality assured) of the "Version 3 Direct Sun" and of the "Version 3.0 Inversions", except for the percentage of sphericity $\%S_{ph}$ for which we used Level 1.5 (in July 2021, Level 2.0 was not yet available for this parameter).



From these datasets, we selected all (1) aerosol thicknesses at 4 wavelengths 440, 650, 850, 1020 nm, (2) aerosol *Ångström*

coefficients between 440 and 850 nm, which allows us to determine the aerosol optical thickness at 550 nm, and (3) microphysical properties $C_{vf}$, $C_{vc}$, $\%C_{vf}$, $\%C_{vc}$, $r_{vf}$, $r_{vc}$, $\sigma_r$, $\sigma_c$, $nr_{440}$, $nr_{650}$, $nr_{850}$, $nr_{1020}$, $ni_{440}$, $ni_{650}$, $ni_{850}$, $ni_{1020}$, $\%S_{ph}$.

A minimum threshold of 50 measurements was used to exclude all sites without a sufficient number of measurements to be representative. As this study was focused on the validation of the atmospheric correction and in an operational context, whereby the atmospheric correction is performed when the aerosol loading is not too high, we decided to limit the data set to aerosol

optical thicknesses at 550 nm of lower than 0.8.

Out of 1139 available AERONET sites, we selected 851 globally distributed sites (Figure 1), resulting in ~1.3 million measurements of aerosol microphysical properties. To characterize the representativeness of these sites, we analyzed the type of land cover surface around the selected AERONET sites. As shown in Figure 2, Urban (24%), Cropland (22%), Forest (17%), Grassland/Shrubland (16%) and Coastal areas and Islands (16%) are more or less equally represented.

For the measurements, AERONET instruments consist of 2 detectors mounted on robots a system developed by *Cimel-France*. One for the measurement of solar (and now lunar) extinction which provides the aerosol optical thicknesses (and then the *Ångström* coefficients) and the water vapor content. The other detector measures the luminance of the day sky using 2 protocols: the almucantar and the principal plane (see Holben et al. 1998, Tables 1 and 2). The almucantar procedure and measurements were used by Dubovik and King (2000) to derive the aerosol microphysical properties. Nevertheless, due to the

protocol of observations, the atmospheric condition (particularly its turbidity and homogeneity), the processing, and the retrieval purpose, the aerosol microphysical properties retrievals are not provided within a single retrieval. There are 3 different sets of retrievals:

(1) the size distribution $C_{vf}$, $C_{vc}$, $\%C_{vf}$, $\%C_{vc}$, $r_{vf}$, $r_{vc}$, $\sigma_r$, $\sigma_c$. This set of parameters is always available when aerosol microphysical properties retrievals are performed by AERONET. For this study, this first block provides a little less than 1.3M

sets of retrievals for the whole 851 AERONET sites used.

(2) the complex refractive index for 4 wavelengths $nr_{440}$, $nr_{650}$, $nr_{850}$, $nr_{1020}$, $ni_{440}$, $ni_{650}$, $ni_{850}$, $ni_{1020}$. This set has a lower occurrence in terms of retrievals, only 0.17M sets of retrievals from 400 sites,

(3) the percentage of sphericity $\%S_{ph}$. This third set is available for the same 851 sites as (1) and provides a little less than 1.3M sets of retrievals. We decided to limit the non-sphericity at a 30% minimum. Indeed, deriving the non-sphericity

integrated over the whole atmospheric column is not unrealistic. Indeed, in almost all cases, particles are randomly oriented. Their integration along the vertical column generates a kind of a minimum sphericity.

The AERONET network has existed since 1993. Figure 3 shows the number of AERONET sites we used for this study during this time period. For the last 9 years, we used more than 350 sites, 250 sites and 350 sites respectively for characterizing the

size distribution, the refractive index and the sphericity. The decrease observed in 2020 is because all data haven't been validated yet.


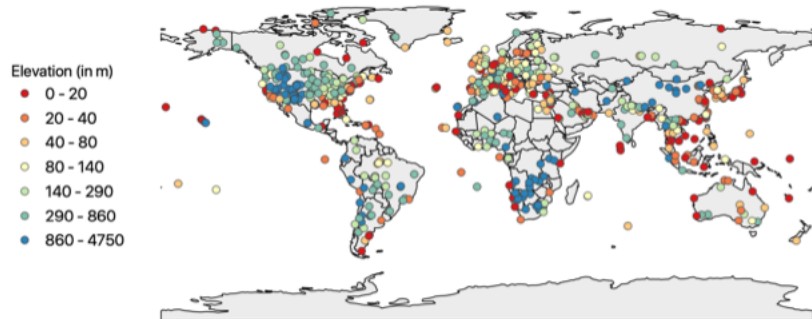

**Figure 1.** Location of the 851 AERONET sites with their elevation.

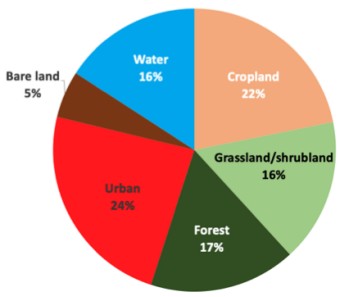


**Figure 2.** Representativeness of land surface types around the selected AERONET sites for the entire selected dataset.

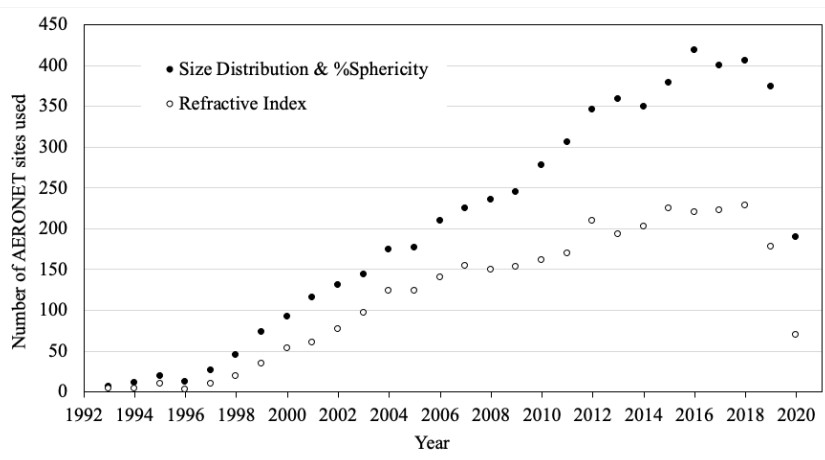

**Figure 3.** Number of AERONET sites selected for this study over the years. The size distribution and the refractive index are level 2.0 while

the sphericity is level 1.5 (see text). The decrease in 2020 is because all data have yet to be validated.





A technical description of values for all aerosol microphysical properties for the 851 AERONET sites is presented Table 1., showing the percentile at 1%, 5%, 95% and 99% and the median value for each of the properties. This gives a global overview of aerosol microphysical properties over land.


**Table 1.** Description of the database of aerosol microphysical properties, for aerosol optical thickness τ at 440 nm, and the Ångström coefficient α (440,870).

| | $\tau_{440}$ | $\alpha_{440-870}$ | $\%$ $C_{vf}$ | $C_{vf}$ $(\mu m^3/\mu m^2)$ | $r_{vf}$ $(\mu m)$ | $\sigma_f$ | $\%$ $C_{vc}$ | $C_{vc}$ $(\mu m^3/\mu m^2)$ | $r_{vc}$ $(\mu m)$ | $\sigma_c$ | $nr_{440}$ | $ni_{440}$ | $\% S_{ph}$ |
|---|---|---|---|---|---|---|---|---|---|---|---|---|---|
| Percentile 0.01 | 0.016 | 0.11 | 5.9 | 0.0020 | 0.093 | 0.34 | 12 | 0.0010 | 1.2 | 0.51 | 1.33 (†) | 0.001 | 30 (*) |
| Percentile 0.05 | 0.031 | 0.28 | 9.3 | 0.0030 | 0.11 | 0.37 | 25 | 0.0040 | 1.4 | 0.55 | 1.36 | 0.002 | 30 (*) |
| Median | 0.14 | 1.26 | 33 | 0.014 | 0.14 | 0.47 | 67 | 0.026 | 2.1 | 0.68 | 1.47 | 0.006 | 63 |
| Percentile 0.95 | 0.62 | 1.85 | 75 | 0.071 | 0.20 | 0.63 | 91 | 0.21 | 3.0 | 0.79 | 1.58 | 0.024 | 99 |
| Percentile 0.99 | 0.89 | 2.03 | 88 | 0.11 | 0.24 | 0.72 | 94 | 0.39 | 3.4 | 0.85 | 1.60 (†) | 0.036 | 99 |

(*) according to our threshold at 30%

(†) according to the AERONET threshold


Each AERONET sites don't have the same number of data (See Figure 4). In the database we developed, one site may contain several thousands of selected retrievals for each aerosol microphysical properties. For example, 8 sites provided more than 10,000 sets of retrievals for the Size Distribution i.e., *Sede Boker* (Israel), *Solar Village* (Saudi Arabia – no longer in the network), *GSFC* (USA), *Burjassot* (Spain), *El_Arenosillo* (Spain), *Carpentras* (France – no longer in the network), *Sevilleta* 150 (USA), *Granada* (Spain). On the other hand, one site may contain less than 100 sets (this is the case for 138 sites). It means that 1 site may represent the equivalent of hundreds of other sites. To avoid the impact of those too well-represented sites, we show in Table 2 the same information as Table 1. By applying a median value per site for each aerosol microphysical retrieval, we have then 686 sets of retrievals. In this case, the median values don't change much (except for $\%S_{ph}$), but the range between both percentiles is reduced, by 20 to 50%. With the assumption of a median value per site, Figure 5 shows the frequency of 155 $\tau_{440}$ and $\alpha_{440-870}$, while Figures 6, 7 and 8 show the frequencies of each aerosol microphysical property from our selected dataset.



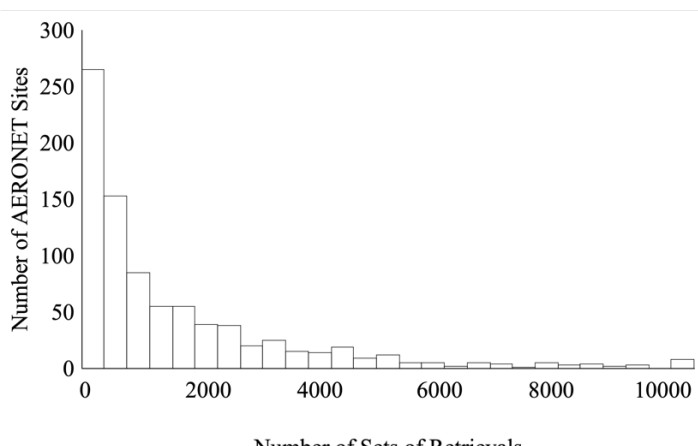

**Figure 4.** Number of sets of retrievals frequency for the aerosol size distribution.


**Table 2.** Same than Table 1, but by affecting median values for each AERONET site.

| | $\tau_{440}$ | $\alpha_{440-870}$ | % $C_{vf}$ | $C_{vf}$ ($\mu m^3/\mu m^2$) | $r_{vf}$ ($\mu m$) | $\sigma_f$ | % $C_{vc}$ | $C_{vc}$ ($\mu m^3/\mu m^2$) | $r_{vc}$ ($\mu m$) | $\sigma_c$ | $nr_{440}$ | $ni_{440}$ | % $S_{ph}$ |
|---|---|---|---|---|---|---|---|---|---|---|---|---|---|
| Percentile 0.01 | 0.031 | 0.34 | 11 | 0.0032 | 0.12 | 0.40 | 24 | 0.0031 | 1.7 | 0.60 | 1.40 | 0.0025 | 30 |
| Percentile 0.05 | 0.066 | 0.55 | 17 | 0.0057 | 0.13 | 0.42 | 36 | 0.010 | 1.8 | 0.62 | 1.42 | 0.0032 | 34 |
| Median | 0.19 | 1.31 | 42 | 0.021 | 0.15 | 0.47 | 58 | 0.031 | 2.2 | 0.67 | 1.47 | 0.0065 | 71 |
| Percentile 0.95 | 0.55 | 1.76 | 64 | 0.065 | 0.17 | 0.55 | 84 | 0.18 | 2.7 | 0.72 | 1.52 | 0.020 | 93 |
| Percentile 0.99 | 0.67 | 1.88 | 76 | 0.088 | 0.19 | 0.60 | 89 | 0.24 | 2.9 | 0.74 | 1.54 | 0.026 | 97 |

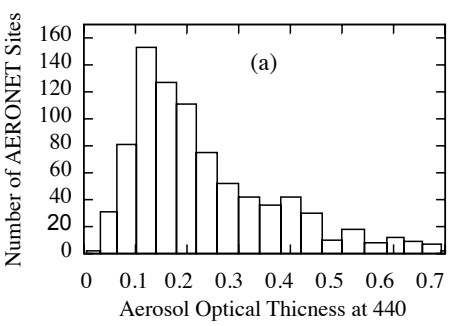
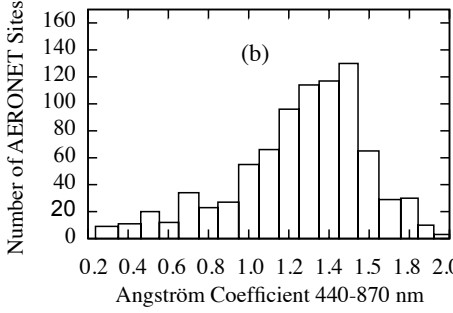

**Figure 5.** Aerosol optical thickness at 440nm frequency (a) and the Ångström coefficient frequency (b).





## 2.3 Metrics used

Results in this paper will be presented in term of accuracy, precision, and uncertainty (APU). If $C_i$ and $R_i$ are the computed and the real values respectively and if N is the number of data, then we can define APU as:

- The accuracy $A$ represents the average bias of the estimates:

$$A = \frac{1}{N}\sum_{i=1}^{N}(C_i - R_i) \tag{2}$$

- The precision $P$ is the deviation around the mean value:

$$P = \sqrt{\frac{1}{N-1}\sum_{i=1}^{N}(C_i - R_i - A)^2} \tag{3}$$

- The uncertainty $U$ encompasses all errors and is derive from A and P

$$U = \sqrt{\frac{1}{N}\sum_{i=1}^{N}(C_i - R_i)^2} = \sqrt{A^2 + \frac{N-1}{N}P^2} \tag{4}$$


The relative uncertainty is defined here as: $U/V$ where $V$ can be the mean value of a specific site or of the whole set of a specific parameter.

## 3 Aerosol microphysical properties

### 3.1 Parameterization of the aerosol microphysical properties

Two protocols of measurements are followed to acquire AERONET data. The Aerosol Optical Thicknesses (AOT) is regularly measured every 15 minutes following a direct measurement of the Sun when cloud-free. For the retrieval of the aerosol model microphysical properties, as specified above, the protocol required an almucantar measurement (Holben et al., 1998; Dubovik and King, 2000), which is realized early in the morning or late afternoon. The main issue is that this AERONET measurement is in general not available during the most common satellites overpass. Moreover, for various reasons (inhomogeneous sky,
small clouds, calibration procedure…) some measurements are missing. We can obviously interpolate data between 2 available measurements, but we can miss the variability of the considered aerosols. Figure 9 shows an example of the impact of changing the aerosol model for size distribution from early morning (7:21:30 local time) to late afternoon (16:28:45 local time). In this example, there is an increase in coarse aerosols between the morning and the evening, but we don't exactly know when that occurred.


In 2002, Dubovik et al. suggested a direct regression (Equation 5) approach versus using the Aerosol Optical Thickness to define each of the microphysical parameters from the AERONET dataset.

Aerosol Microphysical Property $= a + b.\tau$ (5)




For each AERONET site, this approach has been used so far for the official validation of the MODIS and VIIRS surface reflectance products (Vermote et al., 2002; Vermote et al., 2014), for the NASA HLS (Harmonization Landsat-Sentinel) project (Claverie et al., 2018; Vermote et al., 2016), and for the CEOS ESA/NASA ACIX exercise (Doxani et al, 2018). Our objective here is to better account for the temporal and spatial variability of the aerosol microphysical parameters, which can't be only related to the aerosol optical thickness itself. In an operational context, another possible and simple variable available for the aerosol description is the Ångström coefficient $\alpha$ (Ångström, 1929). Indeed, it's well accepted that this coefficient relies to a first order on the aerosol size (which is important in term of light-matter interaction). If we take the example given in Figure 9, we can see from Figure 10 that the aerosol optical thickness doesn't change between the two almucantar procedures, while the Ångström coefficient does. The value of the latter decreases, indicating a bigger particle represented by a bigger coarse mode, which is consistent with Figure 9. Another reason to choose the Ångström coefficient $\alpha$ is mathematical. The aerosol optical thickness $\tau$ is an extensive parameter, the Ångström coefficient $\alpha$ is an intensive parameter, and it's preferable to have a couple of intensive/extensive variables in physical parametrization. Indeed, an intensive parameter can be used for identifying a sample while an extensive parameter can be used for describing this sample.

We decided to select the Ångström coefficient for the 440 and 870 nm wavelengths, i.e., $\alpha_{440\text{-}870}$. Even if the Ångström coefficient has a dynamic behavior over the visible range and it is not entirely constant, $\alpha_{440\text{-}870}$ is a good compromise between all values. At the end, we selected $\tau_{440}$ and $\alpha_{440\text{-}870}$ as variables of the regression. Within the AERONET network, these variables are available every 15 minutes under clear sky condition for all sites.

We can also use the water vapor content, as it's a very important parameter in terms of the microphysical properties. Some aerosols are hydrophile, other are hydrophobic. Water vapor also modifies the size of the aerosol and its absorption capacity. We explored this option, but it didn't improve the retrieval in term of uncertainties. The aerosol optical thickness parameter already includes the effect of the water vapor over the aerosol size distribution, and it explains in part why there were no improvements.

One limited aspect of our approach is that these two parameters $\tau_{440}$ and $\alpha_{440\text{-}870}$ directly correspond to the aerosol scattering, and we may not properly characterize the aerosol absorption (Fraser and Kaufman, 1985; Vermote et al., 2007; Russell et al., 2010; Giles et al., 2012; Lenoble et al., 2013; Tsikerdekis et al., 2021). Therefore, the complexity of the radiative transfer through the atmosphere partially allows mitigation of this phenomenon. Indeed, coupling between the scattering and the absorption of light allows us to indirectly capture the aerosol absorption information.





**Figures 6.** Size distribution parameters frequency for the fine mode (left) and the coarse mode (right).

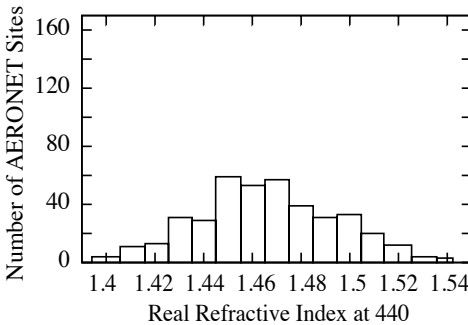
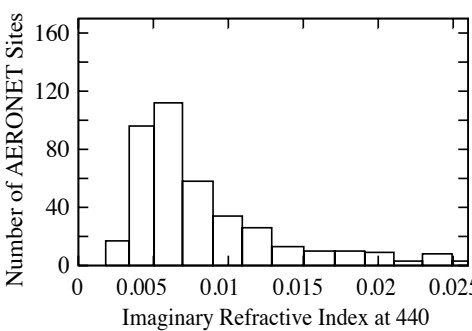

**Figures 7.** Real (left) and imaginary (right) refractive frequency.

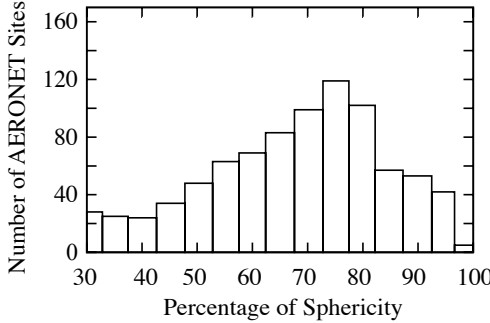

**Figure 8.** Percentage of sphericity frequency.

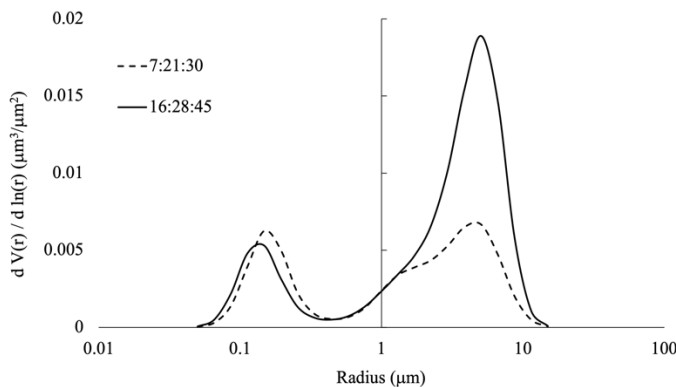

**Figure 9.** Example of an aerosol size-distribution from AERONET with a change between 2 almucantar procedures occurring between the

early morning and late afternoon observations.



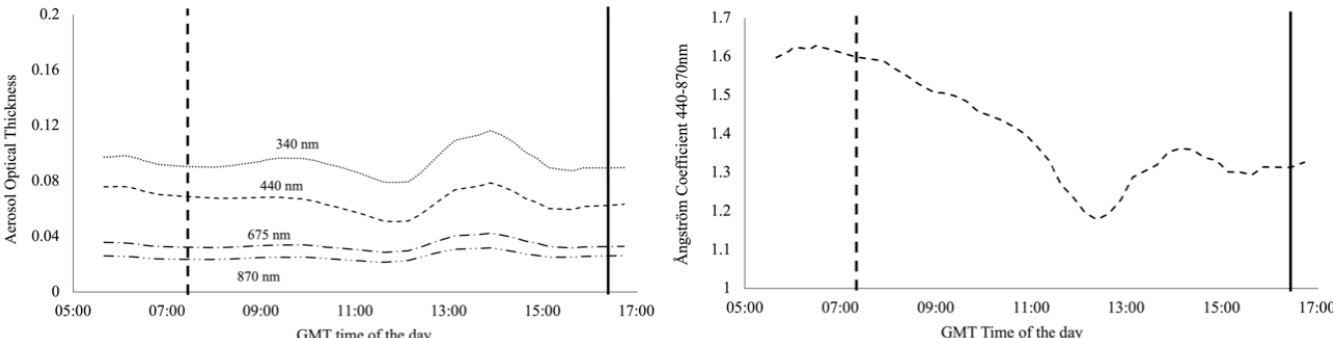

**Figures 10.** Daily variability of the Aerosol Optical Thickness (left) and of the Ångström coefficient 440-870nm (right) for the example of Figure 9.

With our AERONET database (over the 400 sites where we have all microphysical properties), we explored several mathematical formulations for a regression between an aerosol microphysical property and the two variables $\tau_{440}$ and $\alpha_{440-870}$. We used a similar idea after the *Dubovik*'s law (Equation 5), but adding the Ångström coefficient:

$$\text{Aerosol Microphysical Property} = a + b.\ \alpha_{440-870} \tag{6}$$

Then, we tested several mathematical formulations using our two predicted variables, and we found that the following formulation was optimal for describing data:

$$\text{Aerosol Microphysical Property} = (a + b.\tau_{440}{}^{c}).(d + e.\alpha_{440-870}{}^{f}) \tag{7}$$

In practice, to better use Equation 7 and for the stability of retrievals, all 6 coefficients *a, b, c, d, e* and *f* are not derived with a single interaction. The aerosol microphysical parameters meanly depend on $\tau$ or on $\alpha$ (they barely depend on both at the same level). Thus, to get a stable retrieval of the 6 coefficients, we used a "so-called" residue approach by checking which variable ($\tau$ or $\alpha$) is the most representative regarding the behavior of the microphysical parameters. Following this procedure, we apply the first regression law $(a + b.\tau^{c})$ or $(d + e.\alpha^{f})$ to derive $(a, b, c)$ or $(d, e, f)$ respectively. Then, using the remaining residue, we apply the second regression law $(d + e.\alpha^{f})$ or $(a + b.\tau^{c})$ to derive the missing triplet of coefficients. Table 3 shows the percentage of occurrence for $\tau_{440}$ or $\alpha_{440-870}$ as the most representative variable for all microphysical parameters and for all available AERONET sites (see Figure 2).

**Table 3.** Percentage of occurrence for the aerosol optical thickness $\tau_{440}$ and the Ångström coefficient $\alpha_{440-870}$ as the most representative variable for each microphysical parameter.





|  | % $C_{vf}$ | $C_{vf}$ ($\mu m^3/\mu m^2$) | $r_{vf}$ ($\mu m$) | $\sigma_f$ | % $C_{vc}$ | $C_{vc}$ ($\mu m^3/\mu m^2$) | $r_{vc}$ ($\mu m$) | $\sigma_c$ | $nr_{440}$ | $ni_{440}$ | % $S_{ph}$ |
|---|---|---|---|---|---|---|---|---|---|---|---|
| $\tau_{440}$ | 6 | 100 | 50 | 22 | 6 | 79 | 29 | 62 | 39 | 22 | 18 |
| $\alpha_{440-870}$ | 94 | 0.1 | 50 | 78 | 94 | 21 | 71 | 38 | 61 | 78 | 82 |

In Table 3, for 7 of the 11 parameters, $\alpha_{440-870}$ is more correlated with the microphysical parameter than $\tau_{440}$. It confirms that the use of $\alpha$ is pertinent to define these parameters. As expected, $C_{vf}$ and $C_{vc}$ are mostly driven by $\tau_{440}$ (Sinyuk et al, 2020), while %$C_{vf}$ and %$C_{vc}$ are driven by $\alpha_{440-870}$. Parameters $C_{vf}$ and $C_{vc}$, which are extensive parameters, are directly related to the volume loading (mass) of the aerosol, and, in fine, to the number of particles (accumulation of particles). Thus, it's not surprising that $C_{vf}$ is more correlated to $\tau_{440}$ than $C_{vc}$. Indeed, we know that the fine mode optically reacts more efficiently in

the visible light than the coarse mode in terms of extinction (Van der Hulst, 1981), considering that the number of particles present in the fine mode is usually much higher than the number of particles of the coarse mode. By the same reasoning, %$C_{vf}$ and %$C_{vc}$, which are intensive parameters, are not sensitive to accumulation, but rather to the spectral dependency of the aerosol extinction, meaning that %$C_{vf}$ and %$C_{vc}$ are more correlated to $\alpha_{440-870}$. In the AERONET processing, the complex refractive index is applied when the AOT is higher than 0.4 at 440nm. This limits the variability in term of AOT and probably

increases artificially the occurrence for $\alpha_{440-870}$.

We applied our approach for the 3 mathematical formulations given by Equations 5, 6 and 7 over the whole selected dataset and present the results in Table 4.

**Table 4.** Mean relative uncertainties (in percent) for each retrieved aerosol microphysical properties modelled using several mathematical
formulations over the whole dataset.

|  | % $C_{vf}$ | $C_{vf}$ ($\mu m^3/\mu m^2$) | $r_{vf}$ ($\mu m$) | $\sigma_f$ | % $C_{vc}$ | $C_{vc}$ ($\mu m^3/\mu m^2$) | $r_{vc}$ ($\mu m$) | $\sigma_c$ | $nr_{440}$ | $ni_{440}$ | % $S_{ph}$ |
|---|---|---|---|---|---|---|---|---|---|---|---|
| $a+b.\tau_{440}$ | 34.1 | 31.8 | 11.9 | 10.1 | 21.9 | 51.6 | 15.2 | 6.9 | 3.1 | 39.5 | 26.7 |
| $a+b.\alpha_{440-870}$ | 24.3 | 66.0 | 12.0 | 9.2 | 16.1 | 59.4 | 14.5 | 7.0 | 3.1 | 38.4 | 23.6 |
| $(a+b.\tau_{440}{}^c).(d+e.\alpha_{440-870}{}^f)$ | 22.6 | 30.3 | 11.4 | 8.8 | 15.0 | 35.0 | 14.1 | 6.7 | 3.0 | 37.5 | 22.8 |

In terms of accuracy $A$ (Equation 2), results show very low values. Except for $C_{vf}$, $C_{vc}$, and %$C_{vf}$ which present an accuracy up to 2%, accuracies of all other microphysical parameters are below 0.1%. For uncertainty $U$ (Equation 4), the third mathematical formulation gives the overall best results (Table 4). As expected, $\tau_{440}$ better represents $C_{vf}$ while in contrast $\alpha_{440-870}$ better

represents the %$C_{vf}$. Finally, including both variables, we get a non-negligible improvement for both volume concentrations (absolute and relative). For the other microphysical properties, we don't observe much of improvement, but the Equation 7 gives consistently better results. One point to be noted is that all microphysical properties provided by the AERONET network have lower uncertainties than those presented in Table 4 (Dubovik et al., 2000; Sinyuk et al., 2020).



As pointed out, $\%C_{vf}$ and $\%C_{vc}$ globally present a better uncertainty than for $C_{vf}$ and $C_{vc}$, but for exactly 20% of sites the volume concentration of the fine mode $C_{vf}$ is more accurate than the relative volume concentration $\%C_{vf}$ (Figure 11). We are unable to find a clear reason to explain that. The only tiny explanation is that aerosols over these sites present a tendency described by (1) lower concentrations than the average (both fine and coarse modes), meaning relatively low optical thickness, (2) a relative lower Ångström coefficient, and (3) a relative lower absorption. Nevertheless, according to the radiative transfer theory used to define the optical properties (Phase Matrix, Scattering and Absorption coefficients), the phase matrix is normalized at the end. Thus, either the couple of volume concentrations $(C_{vf}, C_{vc})$ or the couple of relative volume concentrations $(\%C_{vf}, \% C_{vc})$ can (it should be a couple) be used depending on the uncertainty for one AERONET site. It should be noted that, in all cases, the uncertainty U of $\%C_{vc}$, $U_{\%Cvc}$, is always lower than that of $C_{vc}$, $U_{Cvc}$.

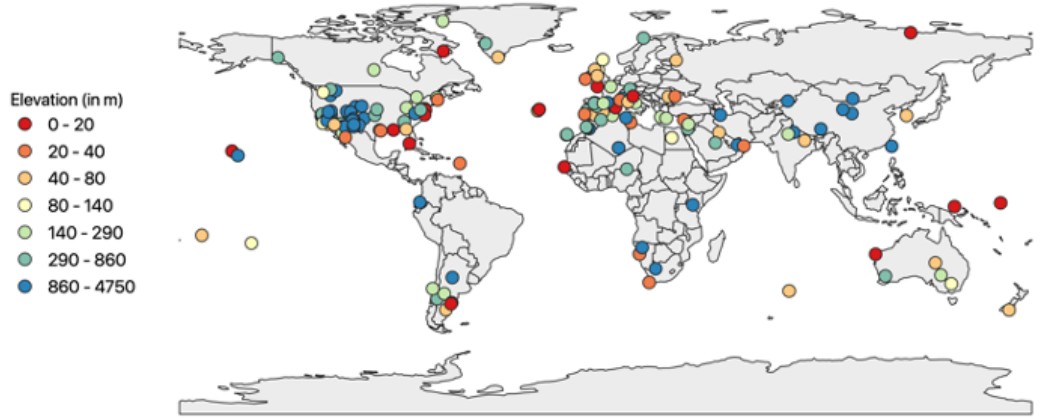

**Figure 11.** AERONET sites for which $C_{vf}$ is better represented than $\%C_{vf}$.

Table 5 shows the new uncertainties $U$ of $\%C_{vf}$, $U_{\%Cvf}$, and the new uncertainties $U$ of $C_{vf}$, $U_{Cvf}$, when we only select sites for which $U_{\%Cvf} > U_{Cvf}$ (80% of cases) or $U_{Cvf} > U_{\%Cvf}$ (20% of cases) respectively. The improvement is visible if we use both $\%C_{vf}$ and $C_{vf}$ according to the lowest uncertainties.

**Table 5.** Uncertainties (in percent) for each retrieved aerosol microphysical properties model (as for Table 4), but after selecting sites for %Cv with $U_{\%Cvf} > U_{Cvf}$ (†, 80% of cases) and for Cv with $U_{Cvf} > U_{\%Cvf.}$ (††: 20% of cases).

| | % $C_{vf}$ | $C_{vf}$ ($\mu m^3/\mu m^2$) | $r_{vf}$ ($\mu m$) | $\sigma_f$ | % $C_{vc}$ | $r_{vc}$ ($\mu m$) | $\sigma_c$ | $nr_{440}$ | $ni_{440}$ | % $S_{ph}$ |
|---|---|---|---|---|---|---|---|---|---|---|
| (a + b.$\tau_{440}{}^c$).(d + e.$\alpha_{440-870}{}^f$) | 22.0 $^{(†)}$ | 22.0 $^{(††)}$ | 11.4 | 8.8 | 15.0 | 14.1 | 6.7 | 3.0 | 37.5 | 22.8 |

As pointed out previously, we have 50% of sites without any refractive indexes. One solution to improve the number of sites is to define mean parameters (a, b, c, d, e, f) for $nr$ and $ni$ by kind of environment (urban, urban coastal, forest, non-forest land,



desert… for example). In that context, we undertook a preliminary study which included all data independently of the site to retrieve mean parameters. It gave a relative uncertainty $U$ of 3.0% for $nr$ with no change compared to Tables 4 and 5. In contrast, for $ni$, it showed a relative uncertainty of 52% for $ni$ which represents about 40% higher than those shown in Tables 4 and 5, but this study includes all data without distinguishing of environment. If we are able to specifically define the environment of the missing sites, we should get a relative uncertainty closer to 37.5% (Tables 4 and 5) rather closer to 52%,

which remains acceptable.

### 3.2 Retrieved microphysical properties from the whole dataset

To expand on Table 4, Figures 12 give the APU of the retrieved microphysical properties over the whole dataset versus $\tau_{440}$ and $\alpha_{440-870}$. The interesting point of these figures is the dependency of uncertainties with $\tau_{440}$ and $\alpha_{440-870}$. Indeed, except for $C_{vf}$ and $C_{vc}$, uncertainties are quite stable with the aerosol optical thickness. On the contrary, most of uncertainties present

variation with the Ångström coefficient. This confirms the importance of considering $\alpha_{440-870}$ in the regression. Another point is the correlation between Tables 4 and 5 and Figures 12. When the variability of the uncertainty with the $\alpha_{440-870}$ is important (Figure 12), the variability of the microphysical properties is more important as well (Tables 4 and 5). It should be noted that for $\%C_{vf}$ and for $C_{vf}$, the APU are for selected sites only (see Table 5).









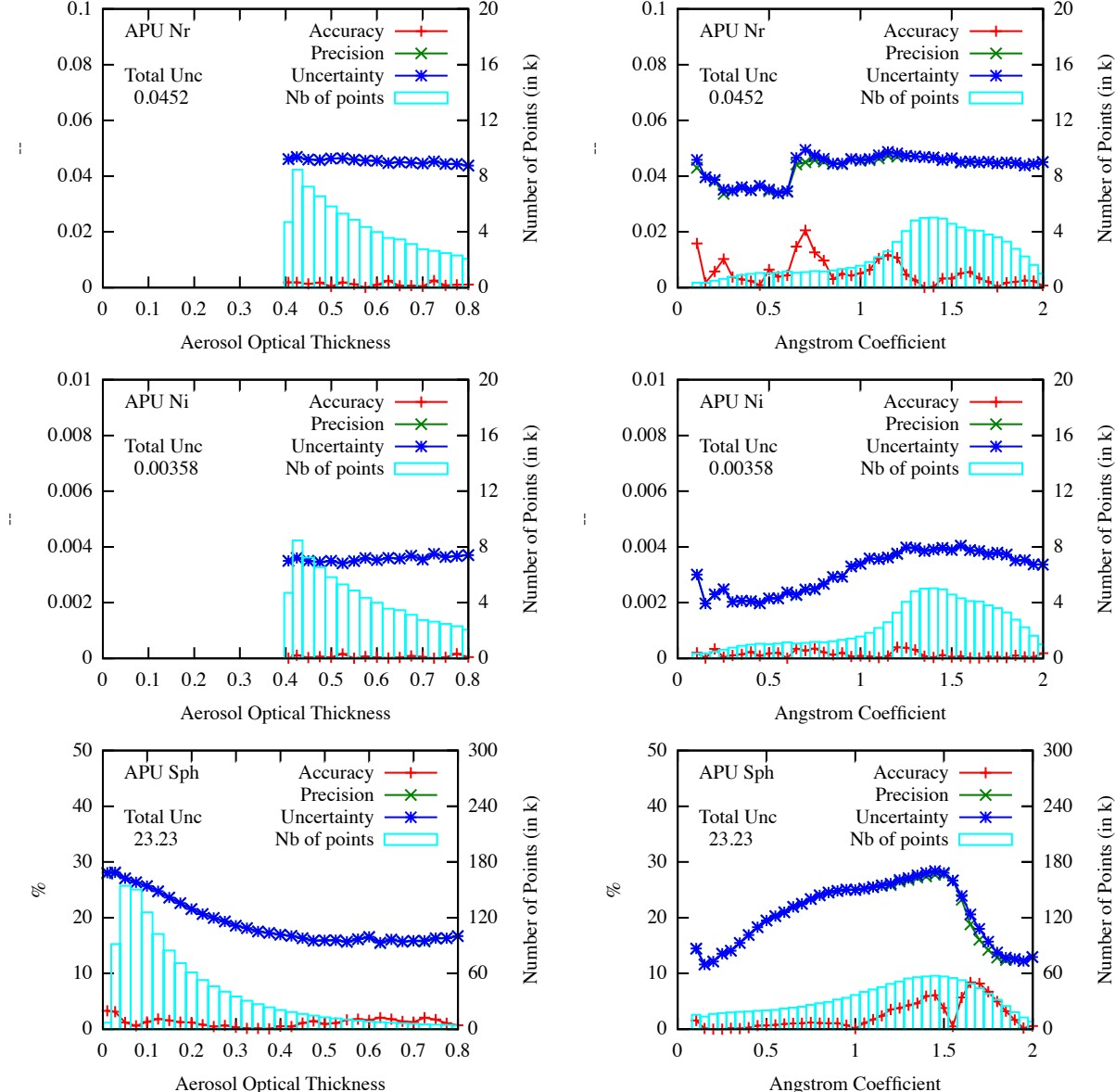

**Figures 12.** APU for the retrieval of each microphysical parameter (from top to bottom: the 8 parameters describing the size distribution - fine and coarse modes, the 2 parameters for the refractive index at 440 nm, and the parameter for the sphericity), versus the aerosol optical thickness at 440 nm (left) and the Ångström coefficient between 440 and 870 nm (right). "Total Unc" represents the total uncertainty of the microphysical parameter.

## 3.3 Retrieved microphysical properties considering each AERONET site

The use of $\alpha_{440-870}$ mostly improves the retrieval of both $\%C_{vf}$ and $\%C_{vc}$ (Tables 4 and 5). Figure 13 shows the comparison between uncertainties on $\%C_{vf}$ and $\%C_{vc}$ using Equation 5 or Equation 7 versus the mean value of $\%C_{vf}$ and $\%C_{vc}$ for each



AERONET site (one dot represents one AERONET site). For $\%C_{vf}$, we only consider sites where $U\%C_{vf} < UC_{vf}$. These figures highlight the improvement of retrievals (about a 1/3 less). We can also point out that relative uncertainties are lower for high and low values of $\%C_{vf}$.

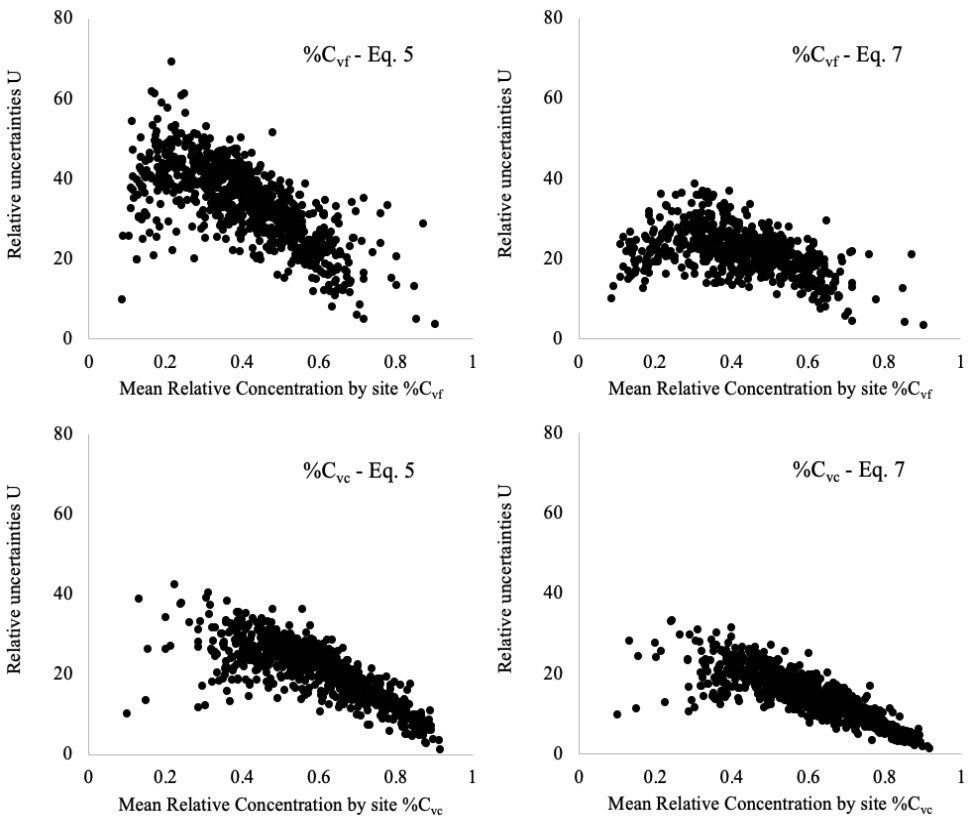

**Figures 13.** Comparison of the relative uncertainties (%) when using Equation 5 (left) and Equation 7 (right) are used to derive $\%C_{vf}$ and
$\%C_{vc}$. One point corresponds to one AERONET site.

Figures 14 give the relative uncertainty for the other microphysical properties site by site, but only using Equation 7 (for $C_{vf}$, we only consider sites where $UC_{vf} < U\%C_{vf}$). Again, except for the volume concentration $C_{vf}$ and $C_{vc}$, we can notice the "arch" effect generating a lower relative uncertainty for lower values and for higher values of the considered properties. It's not shown
here, but this "arch" effect is even more important with absolute uncertainties. At the end, we are able to characterize the uncertainties for each aerosol microphysical property and for each AERONET site.







**Figures 14.** Relative Uncertainties of the aerosol microphysical properties versus the property itself using Equation 7 (one point corresponds
to one AERONET site).

### 3.4 Impact of the uncertainties on the surface reflectance product

As previously mentioned, this work is to support atmospheric correction validation. Thus, one question is: how does the
uncertainty of the retrieved aerosol microphysical property affect the surface reflectance product validation? To address this
issue, we decided to define, for each aerosol microphysical property, the impact of its uncertainty (Table 5) on the atmospheric
correction, and the determination of the surface reflectance. For that purpose, we defined a synthetic database of TOA
reflectances for each AERONET site and for each specific satellite band. To generate this database, we used the 6S code
(Vermote et al., 1997; Kotchenova et al., 2006; Kotchenova et al., 2007; Kotchenova et al., 2008) with the following inputs:
(1) a set of 80 viewing conditions (solar angle, view angle, azimuth angle), (2) a set of different atmospheres (pressure,





temperature, water vapor), (3) a set of surface reflectances (from 0 to 0.6 depending on the wavelength), and (4) a set of 40

aerosol microphysical properties with associated $\tau_{440}$ and $\alpha_{440-870}$ picked up in the real AERONET database. Then, we applied

the atmospheric scheme developed for the Land Surface Reflectance Code (LaSRC) algorithm for MODIS, VIIRS, Landsat-8, Sentinel-2 (Vermote et al., 2002; Vermote et al., 2014; Vermote et al., 2016; Claverie et al., 2018; Doxani et al., 2018). First, using each set of input, we computed the TOA reflectance. Then, inducing 20 cases of random uncertainties for each aerosol microphysical properties, we applied an atmospheric correction to get the surface reflectance $\rho_{surf}$ to be compared to

the one used as input. Table 6 gives the uncertainties we get for the MODIS red channel (band 1, 620-670 nm). The main relative uncertainty appears for the uncertainty $Uni_{440}$ of the imaginary part of the refractive index (relies to the aerosol absorption), $1.0 \ 10^{-3}$ in terms of surface reflectance, followed by the uncertainty of the radius of the fine mode. In a decreasing order of magnitude, $Ur_{vf}$ and $Unr_{440}$ appear around a third lower. Then, another step below, appears $UC_{vf}$ and $U\% \ C_{vf}$.

**Table 6.** Surface reflectance uncertainties (for the MODIS Red channel) due to the initial aerosol model uncertainties.

| | $\% \ C_{vf}$ | $C_{vf}$ | $r_{vf}$ | $\sigma_f$ | $\% \ C_{vc}$ | $r_{vc}$ | $\sigma_c$ | $nr_{440}$ | $ni_{440}$ | $\% \ S_{ph}$ |
|---|---|---|---|---|---|---|---|---|---|---|
| Initial relative Uncertainty (Table 5) | 22.0 % | 22.0 % | 11.4 % | 8.8 % | 15.0 % | 14.1 % | 6.7 % | 3.0 % | 37.5 % | 22.8 % |
| In-fine uncertainties on surface reflectances | $1.4 \ 10^{-04}$ | $1.5 \ 10^{-04}$ | $3.9 \ 10^{-04}$ | $4.0 \ 10^{-05}$ | $6.7 \ 10^{-05}$ | $5.5 \ 10^{-05}$ | $2.6 \ 10^{-05}$ | $3.6 \ 10^{-04}$ | $1.0 \ 10^{-03}$ | $6.0 \ 10^{-05}$ |

Many atmospheric correction schemes use a blue channel to retrieve the aerosol properties, so it's interesting to assess the impact of the aerosol model with the atmospheric reflectance in the blue channel. Figure 15 shows, for an example with the MODIS blue channel (band 3), the dependency between the uncertainties on $\rho_{surf}$ in the red channel and the atmospheric

reflectance in the blue channel. This uncertainty is always low, below 0.005, for a range of reasonable atmospheric reflectance values. This figure also shows that this atmospheric reflectance in the blue channel is almost linearly correlated to the uncertainties of the surface reflectance in the red channel. This means that a QA flag can be directly defined using the atmospheric reflectance in the blue channel rather than the optical thickness (Vermote et al., 2002; Vermote et al., 2014).





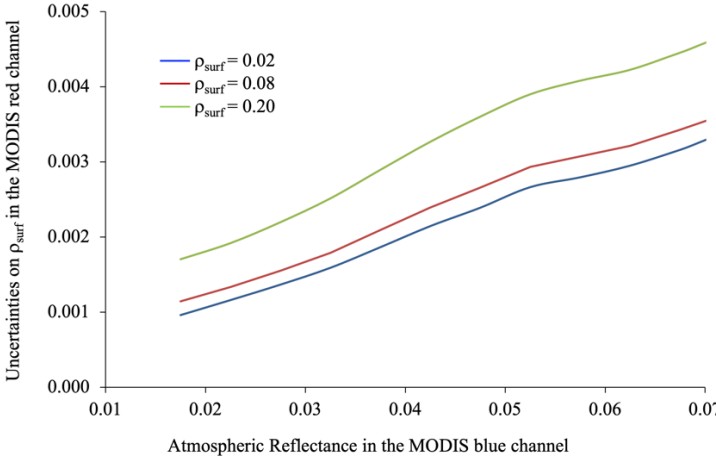


**Figure 15.** Uncertainties of $\rho_{surf}$ in the MODIS red channel versus the atmospheric reflectance in the MODIS blue.

Finally, Figure 16 represents, in fine, the impact of the aerosol model uncertainties retrieved using Equation 7 on the surface reflectance retrieval $\rho_{surf}$ in the MODIS red spectral band. Uncertainties, shown for two ranges of aerosol optical thicknesses

at 550nm $\tau_{550}$ - less than 0.25 and less than 0.50, are clearly always below the MODIS specification required for the surface reflectance ($0.005 + 0.05*\rho_{surf}$). For $\rho_{surf}$ ranged between 0.10 and 0.40, the uncertainty on $\rho_{surf}$ is relatively between 1 and 2%. It confirms that our aerosol model description for the AERONET sites can be used with a good confidence for the satellite atmospheric correction.

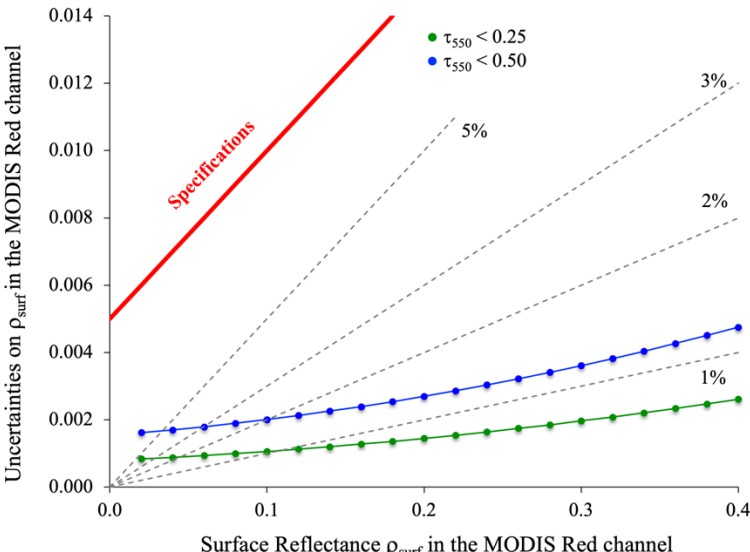

**Figure 16.** Uncertainties of $\rho_{surf}$ versus $\rho_{surf}$ in the MODIS red channel. Relative uncertainties on $\rho_{surf}$ (1, 2, 3 and 5%) are highlighted.





## 4 Conclusion

This study was aimed at defining and building an aerosol model based on the microphysical properties obtained for 851 AERONET sites. The objective is to use this characterization for validation of atmospheric correction of space-borne remote sensing sensors, but this can be extended to many other uses. Using the optical thickness at 440 nm $\tau_{440}$ and the Ångström

coefficients $\alpha_{440-870}$ of aerosols, we characterized the microphysical properties of aerosols with an acceptable uncertainty (from 6.6% to 20.7%), the imaginary part of the refractive index being the least well-rendered parameter (c. less than 40%), which is not a surprise since this parameter is the most difficult to retrieve from optical measurements. The study shows different behaviors according to the value of each microphysical property, showing an arch effect resulting from lower uncertainty for the highest values and the lowest values of the microphysical property.

In terms of atmospheric correction, this method can be used to define a surface reflectance reference as we do for the validation of surface reflectance products for sensors such as MODIS, VIIRS, Landsat, Sentinel. An impact study of the uncertainties of each microphysical property of the aerosols showed that the aerosol models used to define a reference surface reflectance provide a maximum uncertainty of 1 to 3 %, well below the specifications often used for atmospheric correction. Nevertheless, it will be important to further test these findings using additional datasets for validation (number of sites and number of

comparisons.

*Data availability*: Data are currently available in our web site (https://salsa.umd.edu). *NOTE TO REVIEWERS: after publication*

*Financial support:* This research was funded by the NASA grant numbers 80NNX17AJ63A and 80NNSC19M0222.

*Acknowledgments:* We thank all AERONET PI investigators and their staff for establishing and maintaining all sites used in

this investigation.

*Competing interests:* The authors declare that they have no conflict of interest.

### *Annex: Nonparametric model approach*

To test the ability of the optical thickness and the Ångström coefficient to be reliable for reproducing the aerosol models, we used a nonparametric approach. A Random Forest (RF) regression model was built with AOT and angstrom coefficient as

inputs and all other parameters as outputs (dependent variables). The data were randomly split into training (50%) and test sets. The split was done in order to analyze the robustness of the model. Performance of the model (APU diagram) was assessed on testing data. The RF model had 100 trees and maximum depth of trees was limited to 15 to avoid overfitting.

Figures A1 give examples of results of this nonparametric approach (for parameters describing the fine mode of the size distribution only, but the conclusion can be generalized to all microphysical parameters). Comparing to Figures 12, we have

similar results for presented examples of retrieved microphysical properties. This indicates that the use of the optical thickness $\tau_{440}$ and the Ångström coefficient $\alpha_{440-870}$ is consistent.




**Figure A1.** APU for each microphysical parameter (fine mode of the size-distribution only) retrieved from a Random Forrest approach versus the aerosol optical thickness at 440 nm (left) and the Angstrom coefficient between 440 and 870 nm (right).



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
