# Peer review of "Aerosol models from the AERONET data base: application to surface reflectance validation"

_Atmospheric Measurement Techniques, 2021_

## Author Response (AR1)

**ANSWERS to REVIEWER 1:**

Thank you very much for all your constructive comments. We mostly agree with all of them. Below is the detail of each answer.

**Page 1:**
**Line 19: "evaluate our own empirically retrieved microphysical properties".**
Thank you, modified.

**Line 22: "…from 3.5X10^-5 to 10^-3 in reflectance units."**
Thank you, modified.

**Line 28-30: There are too many citations here. Include only important ones.**
OK.

**Page 2:**
**Line 33-35: put this sentence outside the bracket.**
Thank you, modified.

**Line 51-53: But this is what is being performed for operational surface reflectance products from MODIS & VIIRS. Isn't it?**
Yes, it does, but the aerosol model we use is simpler.

**Line 54: "Numerous studies have shown the validation of aerosol optical depth products derived from various sensors, i.e., MODIS, MISR, OMI, POLDER, and Landsat.**
Thank you, modified.

**Line 60: These are aerosol models describing their optical-microphysical properties (not just optical or microphysical)**
You're right, it's a typo mistake, I corrected it.

**Page 3:**
**Line 85: Non-sphericity mostly applies to coarse mode dust particles; fine mode aerosols are adequately modeled as spherical particles.**
Yes, and if you do not use polarization, then using assumption of spherical particles is ok for most applications. Even with polarization the difference of scattering between spherical and non-spherical particles for small sizes is not really significant. So, we defined this parameter in case users will need it.

**Line 88: "AERONET measurements".**
Thank you, modified.

**Page 4:**
**Line 92: "aerosol optical depths"**
Thank you, modified with thickness.

**Line 93: "860 nm"**
Yes, there is a typo mistake. In reality, it's 675 and 870 nm

**Line 95: 50 measurements in direct product or inversion product?**
Inversion, I rewrote the sentence.

**Page 5:**
**Figure 1: It would be more informative to color-code each site with its corresponding length of measurements (in years).**
Yes, that's a good idea. Nevertheless, the length of measurement is not totally indicative as we can have several months without data (it happens when there is maintenance or for re-calibration). We decided de re-do Figures 1 and 11 with the number of data rather than elevation.

**Figure 1 and Figure 3 can be combined in one plot shown panels side-by-side.**
Please, see the comment about Figure 1.

**Page 7: It is not understood how Table 2 was derived. Please explain the methodology clearly.**
Thank you, we re-wrote the methodology and the text describing Table 2. To explain: We derive 1 median value (for each microphysical parameter) for each site, and then we do the median of the 851 sites (for each microphysical parameter). It avoids the too important weight of sites with numerous data (some sites provide 100 data when others provide 10,000).

**Page 8: Section 2.3 Metrics used: This section is not adequately described.**
I understand you comment, but I do not know what to say exactly. The APU are more and more used in our AC community (it has been adopted by the CEOS ESA-NASA ACIX exercise), and there are now many papers with APU. So, I tried to rewrite the first sentence of chapter 2.3. I hope the text is clearer.

**Page 9:**
**Line 185: From which AERONET site this data comes from? What is the time of the year? It also applies to Figure 10.**
Yes, you're right. I added the origin of data in both Figures 9 and 10 caption.

**Line 190: A brief description of how such regression was derived is required here.**
I rewrote the sentence by more precise (and trying to remain simple).

**Line 213: "hydrophilic".**
Thank you, modified.

**Page 12:**
**Line 234-235: This isn't completely true as total AOT is comprised of scattering and absorption AOTs.**
I'm sorry, I don't see what you are talking about, your lines do not correspond to the initial version. Could you precise please?

**Line 236-238: This isn't understood. Please clarify.**
I'm sorry, I don't see what you are talking about, your lines do not correspond to the initial version. Could you precise please?

**Eq. 7: Why AOT of coarse mode and AE of fine mode are used here? Can author use total AOT & AE instead? How do numbers in Table 4 (last row) change If total quantities in both parameters used?**
I think there is a misunderstand, there are 6 regression parameters a, b, c, d, e, f and c and f don't stand for coarse and fine.

**Page 20:**
**Section 3.4, line 350-355: This is essentially a look-up table of TOA reflectance under varying atmospheric and surface conditions.**
Yes, we can say that.

**Page 21:**
**Table 6: What is "In-fine uncertainties"? Also, uncertainties in MODIS blue channel (band 3) should be included in Table 6.**
This is the uncertainty generated at the end on the surface reflectance product once we apply the uncertainty on the microphysical parameter. For example, % Cvf is generated with an uncertainty of 22.0%. This uncertainty generates, once we proceed an atmospheric correction scheme, an uncertainty of 0.00014 on the surface reflectance (in reflectance unit). I add explanation in the text.

**Figure 15 should be plotted as uncertainties in surface reflectance in blue channel (y-axis) versus that in red channel (x-axis).**
I don't understand. Most of the atmospheric correction procedures over land (I added this point in the whole paper – "over land") use blue channels to retrieve the aerosol information as the surface reflectance is very low over most of lands in the blue. For that purpose, the atmospheric reflectance is usually uses. So, Figure 15 shows how uncertainties due to the use of our aerosol models is reported on the surface reflectance in the red channel (which is the most used).

**Page 22:**
**What is difference between results shown in Figure 16 and tabulated in Table 6? The figure caption is written poorly. Please explain in the caption what each line and dots represent.**
Figure 16 includes all uncertainties (derived from each microphysical parameter) while Table 6 reports individual uncertainties. I explained in detail the figure in the caption.

**Line 389: The aerosol model is developed using optical properties (AOT and AE), not microphysical properties, where the latter is actually derived from the former two, as stated on line 391-392.**
AOT and AE are used to generate the microphysical parameters. Once we get these parameters, we recompute the optical properties. I collaborate regularly with people working on the aerosol chemistry. For them, an aerosol model is defined by their microphysical parameters. I agree with that view as optical properties are derived from microphysical properties.

**Line 391: "…many other applications."**
Thank you, modified.

**Conclusion should be expanded and adequately discuss the methods adopted and results obtained in this work.**
We expanded the conclusion adding details about the adopted method and results.

**ANSWERS to REVIEWER 2:**

Thank you very much for your comments and the time you take for the review. We tried to intercorporate all your suggestions. Please, see below for the detail of each comment and question.

**General comments:**

**The objective of the manuscript is somewhat confusing. The title suggest "the use of aerosol models for the validation of surface reflectance", the abstract suggests that the objective concerns building and identifying aerosol models. Finally, the introduction suggests that the objective is the description of these "dynamic" aerosol models definition and design.**
Right, it might have a confusion. The abstract and the introduction have been rewritten to avoid as much as possible the confusion. The first goal of the paper is the creation of aerosol models by describing their microphysical properties for the each AERONET sites. The second part is the evaluation of uncertainties using our microphysical properties when we build a surface reflectance reference to be used for the validation of satellite surface reflectance products over land.

**The introduction does not reflect well enough the paper objective and structure. Please define clearly the objective within the paper title, abstract and introduction. Detail how the paper is organised to reach the proposed objective(s).**
As mentioned above, the abstract and the introduction have been rewritten to clearly state the objectives of the paper

**In the context of the radiative transfer theory used here to perform atmospheric correction, the authors do not justify the choice of the proposed strategy. Single scattering (or optical, ie, single scattering albedo and phase function) properties directly impact the propagation of light in the atmosphere following this theory. Different combinations of micro-physical properties might lead to similar optical properties. There is therefore no need to develop such kind of aerosol model based on micro-physical properties. Starting from the optical properties is simpler and leads to less possible confusion. Please compare these two approaches and justify the proposed approach.**
Yes, optical properties drive the propagation of light through the atmosphere, but the optical properties are derived from the microphysical properties which are derived from the aerosol composition (and the aerosol chemical component). From the origin of the aerosol transfer radiative, aerosol models have been described by their microphysical properties, or their optical properties, or both. There is no real confusion, but we add sentence to explain (briefly) our choice. Indeed, rather than using "mean" values for the scattering coefficient, the absorption coefficient, and the phase matrix (because we need the matrix to consider the polarization which plays a nonnegligible role in the blue range of the solar spectrum), we prefer to use "mean" value of the microphysical properties and then derive the optical properties. It gives the user the choice of what he needs. And, sometime, people use SSA and the asymmetry parameter. Thus, our choice allows users to do everything they want, or they need.
Another point is the loss of the aerosol information when we provide directly optical properties. Microphysical properties give information on the aerosol structure and, in a frame of a general use of the aerosol model we provide, it's important to let all information available to the user.
Last point (not mention in the paper), providing optical properties means providing, at least (1) the scattering and absorption coefficients for all wavelengths, and (2) the Phase Matrix for all scattering angles. It's not so simpler in term of data managing.

**The selection of the experimental setup used in Sections 3.4 is not discussed at all. It is also not clear how t_440 and alpha_440-870 can be derived to use the proposed approach. The benefit of this method is therefore not clearly demonstrated. Consequently, the approach proposed in this paper appears pretty much irrelevant as can be seen from the absence or convincing conclusions.**
AOT (and alpha_440-870) is the most available aerosol parameter. If we don't have it, we do almost nothing. So, it's not irrelevant at all!!
As it's said in the paper, we used this approach with the 2002 Dubovik's model which only use AOT, and, so far, it's one of the most cited papers and no one considers this approach as irrelevant. Here, we improved what Dubovik did. For several years, we use the one presented in the paper. And when we perform the validation for one specific satellite, we do have thousands and thousands

(sometime hundreds of thousands) point of validation… so it's relevant. (see below for some detail of how we do – point 21).

**Detailed comments**

1. **Abstract: Second sentence "As part of the validation of atmospheric correction of remote sensing data affected by the atmosphere, it is critical to utilize appropriate aerosol models as aerosols are a main source of error" Is the aerosol model more important to characterise than the aerosol optical thickness?**
   The AOT is important for sure because it drives the whole amplitude of the aerosol atmospheric signal. Nevertheless, the aerosol model is important as well. The way scattering and absorption are characterized modifies the aerosol transmission and the aerosol atmospheric reflectance. Moreover, we talk about validation, meaning all aerosol information are essential. You can do atmospheric correction with the simple aerosol model if you want, but for validation, we have to be as realistic as possible with the local environment (of the place of validation). This has been demonstrated and published (at least by Justice et al, 2005).

2. **Abstract third sentence "In this paper, we propose and demonstrate a framework for building and identifying an aerosol model". This sentence is not clear. What is the purpose?**
   The abstract has been rewritten.

3. **Abstract: last two sentences. Uncertainties are given in absolute reflectance units and relative. Please provide uncertainties in a coherent way throughout the manuscript.**
   For both results, absolute uncertainties in reflectance unit are now present in the new version.

4. **introduction, first sentence: It is written … properties for a better evaluation of their impacts. Aerosol impact on what? Please clarify.**
   The introduction has been rewritten.

5. **Line 67: It is written: the complex refractive index (gives the path of light through the atmosphere). This statement is inaccurate. The path of light also depends on the radius. Please correct this statement.**
   It has been rewritten.

6. **Line 70-71. It is written "For an optical approach, the use of the Gaussian Distribution is widely accepted". Please add a reference.**
   That's a consensus, there are thousands of people saying it, but no real reference. We added the 2 first publications which cited it (whitby, 1978 ; Shettle and Fenn, 1979).

7. **Line 89. Are these references correct for version 3? They look a bit old for version 3.**
   Yes, there is one about V3. We added it in the updated version.

8. **Line 96 and 116: Some symbols are not defined or not consistent with previous definition. Please define and use symbols consistently throughout the manuscript.**
   Sorry, in the version we've got, we don't see what you are referred to? Could please be more precise? Maybe, you are talking about Equation 1 as there was missing indexes in this equation. We already corrected them. We also double check all the manuscript in the updated version.

9. **Line 98: It is written "As this study was focused on the validation of the atmospheric correction and in an operational context …" Is this yet a new or different objective of this manuscript?**
   The paragraph was not clear enough, it has been rewritten.

10. **Line 120: If it not unrealistic, does that mean it is realistic?**
    +
    **Line 121: can you please elaborate this sentence: "Their integration along the vertical column generates a kind of a minimum sphericity."**
    This paragraph has been rewritten.

**11. Line 124: "this time period" do you mean the last 9 years?**
No, since 1993. It's now better explained in the new version.

**12. Figure 4: The X axis title and figure legend are misleading. The axis title suggests that all data sets are considered whereas the legend suggest that only the first one is used.**
It has been corrected.

**13. Line 166: The definition of symbols C_i is not clear. Please specify the nature of the computed values.**
Ci is the computed value. Sorry, we thought it was clear enough, but we detailed a little more its definition in the new version.

**14. Line 207: please clarify and justify the statement made in this sentence : "Indeed, an intensive parameter can be used for identifying a sample while an extensive parameter can be used for describing this sample".**
 That's the definition of an intensive or an extensive parameter.

**15. Line 209: It is written "We decided to select the Ångström coefficient for the 440 and 870 nm …" Why did you take this decision? Please clarify.**
+
**Line 211: "At the end, we selected t440 and a440-870 as variables of the regression" You mean Equation 5? Could you please justify this choice.**
Equation 5 is what Dubovik suggested and only relies to tau440. The whole paragraph written above explain why it's worth to use a440-870. The discussion of the law we choose is done below in the chapter.

**16. Equation 6 and 7. Please use symbols for the left side of these equations**
It was written in that way to simplify the reading

**17. Lines 254, 257, 258: please use the correct symbols for alpha and tau. Such loose usage of symbols occurs elsewhere but will not be reported in this review.**
Not sure to understand what you said as we use the correct symbol. Just in case, we modify tau by tau_440 and alpha by alpha_440-870 (we sometime don't write 440 and 440-870 because we didn't want to overload the reading, but well it has been added).

**18. Line 259: Please be more rigorous in the definition of the most representative variables.**
We agreed it was not clear, it has been modified

**19. Figure 12: left axis is title is not readable**
It has been added

**20. Line 352: it is not clear how this work is used in practice to support atmospheric correction validation.**
-We rewrote the introduction to better explain it. To define a surface reflectance as reference, we need to know, at least, the aerosol reflectance, the aerosol transmittances and the atmospheric spherical albedo. For that, we need the aerosol optical properties which, in our case, are derived from our aerosol microphysical properties.
-When we perform an atmospheric correction validation, we look at all AERONET sites, when we found for a site an AOT available at the time of the satellite overpassing, we perform a validation. If we have AOT, we have the Angstrom coefficient, then we can derive (with our methodology) the microphysical properties, then the optical properties, and then the aerosol atmospheric reflectance, the aerosol transmittances and the atmospheric spherical albedo.

**21. Figure 15: which angular configuration is used? Why the magnitude of the x axis stops at 0.07? Is the surface assumed Lambertian? Please justify the experimental setup.**
-Inputs are described in the first chapter of 3.4. The angular configuration describes the whole satellite angular configuration possible. It has been added in the new version
-When we define a surface reflectance reference, yes the surface is Lambertian.
-The x axis stops at 0.07 because, beyond that, it starts to do not make sense as the aerosol loading is too high to be corrected properly. For example, in our scheme of atmospheric

correction scheme, one of the criteria to perform atmospheric correction over a pixel is to have the aerosol atmospheric reflectance in the blue lower than 0.03.

**22. Figure 15: Please add the requirements**
Which requirements are you talking about for Figure 15??

**23. Figure 16: Please provide a reference for the requirement definition.**
The MODIS specification has been defined by the Science Team, but there is no reference.

**24. Line 400: Could you please clarify how this method can be use to define a surface reflectance reference?**
See point 22